# A Narrative Review on Means to Promote Oxygenation and Angiogenesis in Oral Wound Healing

**DOI:** 10.3390/bioengineering9110636

**Published:** 2022-11-02

**Authors:** Wei Cheong Ngeow, Chuey Chuan Tan, Yet Ching Goh, Tatiana Miranda Deliberador, Chia Wei Cheah

**Affiliations:** 1Faculty of Dentistry, University of Malaya, Kuala Lumpur 50603, Malaysia; 2Latin American Institute of Dental Research and Education—ILAPEO, Curitiba 80710-150, Brazil

**Keywords:** oxygen, wound, oral mucosa, angiogenesis, healing

## Abstract

Oral mucosa serves as the primary barrier against pathogen invasions, mechanical stresses, and physical trauma. Although it is generally composed of keratinocytes and held in place by desmosomes, it shows variation in tissue elasticity and surface keratinization at different sites of the oral cavity. Wound healing undergoes four stages of tissue change sequences, namely haemostasis, inflammation, proliferation, and remodelling. The wound healing of oral hard tissue and soft tissue is largely dependent on the inflammatory response and vascular response, which are the targets of many research. Because of a less-robust inflammatory response, favourable saliva properties, a unique oral environment, and the presence of mesenchymal stem cells, oral wounds are reported to demonstrate rapid healing, less scar formation, and fewer inflammatory reactions. However, delayed oral wound healing is a major concern in certain populations with autoimmune disorders or underlying medical issues, or those subjected to surgically inflicted injuries. Various means of approach have been adopted to improve wound tissue proliferation without causing excessive scarring. This narrative review reappraises the current literature on the use of light, sound, mechanical, biological, and chemical means to enhance oxygen delivery to wounds. The current literature includes the use of hyperbaric oxygen and topical oxygen therapy, ultrasounds, lasers, platelet-rich plasma (PRP)/platelet-rich fibrin (PRF), and various chemical agents such as hyaluronic acid, astaxanthin, and *Centella asiatica* to promote angiogenesis in oral wound healing during the proliferation process. The arrival of a proprietary oral gel that is reported to improve oxygenation is highlighted.

## 1. Introduction

Oral mucosa provides a delicate lining for the oral cavity and serves as the primary barrier against pathogen invasions, mechanical stresses, and physical trauma. Oral mucosa displays architectural similarities to skin, having a superficial epithelium that is composed of keratinocytes and held in place by desmosomes (Figure 1). Distinctive structural and functional differences can be observed between these two types of tissues. Oral mucosa shows variation in tissue elasticity and surface keratinization in different sites of the oral cavity [1]. The gingival and palatal regions are lined with an epithelium with an increased surface keratinization to withstand greater mechanical forces, whereas the buccal mucosa is more elastic and loosely arranged [1]. In the occurrence of an injury, oral wounds reportedly demonstrate rapid healing, less scar formation, and fewer inflammatory reactions compared to skin wounds [1,2]. This privileged healing of oral wounds is associated with a less-robust inflammatory response, favourable saliva properties, a unique oral environment, and the presence of mesenchymal stem cells [3].

## 2. Oral Wounds

Oral wounds can involve oral soft tissue and/or hard tissue. Oral wounds may arise following chemotherapy or radiotherapy, mucocutaneous disorders, trauma, extractions, dental implants, or other invasive dental procedures [3,4]. Oral mucositis in patients receiving chemotherapy or radiotherapy is particularly painful, leading to the deterioration of their quality of life [4]. The clinical course of oral mucositis is significantly associated with the dose of radiation administered and the selection of stomatotoxic agents in chemotherapy [5]. Even a non-medically inflicted wound, such as an aphthous ulceration, is a major concern to the patients and those treating these patients. Some of these ulcers just refuse to heal or keep recurring following remission.

Following trauma or an ulceration, wound healing is initiated and four stages of tissue change sequences, namely haemostasis, inflammation, proliferation, and remodelling, take place [1,3]. Haemostasis and inflammation will start from the moment of injury and continue for up to 4 to 6 days. The proliferation stage involves re-epithelialization, angiogenesis, granulation tissue formation, and collagen deposition. In hard-tissue healing, there is an additional mineralization stage. This phase will take place starting from day 4 and last up to three weeks following a soft-tissue injury. The remodelling phase of both soft and/or hard tissues will follow and proceed for about one year [1,3] (Figure 2). The wound healing of oral hard tissue and soft tissue are largely dependent on the inflammatory response and vascular response; the latter is the focus of the current review paper. 

### Angiogenesis in Oral Wound Healing

Angiogenesis, or neovascularization, is the hallmark process and plays a critical role in wound healing [6]. This process includes the reestablishment of the existing vascular network and production of a new dense, but loosely arranged, capillary bed [7,8,9]. High capillary growth is essential for optimal wound healing because it provides oxygen and micronutrients and removes catabolic waste products from the healing tissues [6]. Angiogenesis is a dynamic interaction among vascular endothelial cells, angiogenic cytokines, and the extracellular matrix microenvironment [8]. Fibroblast growth factor (FGF), vascular endothelial growth factor (VEGF), angiogenin, transforming growth factor (TGF-β), angiopoietin, and mast cell tryptase are among the angiogenic mediators reported [1,8,10]. Blood vessels can constitute up to 60% of the granulation tissue in healing wounds [8]. As the extracellular matrix undergoes maturation, blood vessel formation will reduce [4]. Diabetes patients are more likely to have disturbed wound healing owing to their underlying immunologic aberrancies and angiogenesis deficiency [4,9]. While angiogenesis is necessary to bring nutrients and oxygen to healing wounds, the presumed functional importance of an overabundance of angiogenesis has recently been challenged. Currently, the oral mucosa is believed to heal with a reduced angiogenic burst composed of more mature vessels that provide better oxygenation [9]. Oxygen plays an important role in wound healing, as it is vital for energy production and protein synthesis, cellular proliferation, angiogenesis, and the restoration of tissue functions. Oxygen levels vary depending on the anatomical location, with the oral cavity deemed as having good blood flow and a high tissue metabolic rate. A wound, however, has a hypoxic environment that is associated with compromised healing, thus increasing the risk of infection [4,11]. However, during the initial inflammatory process of wound healing, this acute hypoxic environment enhances fibroblast proliferation and alters normal stromal cell function. Hypoxic conditions induce fibroblasts to increase the secretion of transforming growth factor (TGF-β1), and subsequently upregulate the expression of hypoxia-inducible transcription factor 1 (HIF-1). HIF-1 acts as the main regulator for oxygen homeostasis, and is an important determinant for cell survival and wound-healing outcomes. HIF-1 is involved in most steps of the healing process, including cell migration, cell division, the release of growth factors, angiogenesis, and extracellular matrix metabolism. The activation of HIF-1 in hypoxic conditions stimulates angiogenic factors such as vascular endothelial growth factor (VEGF), matrix metalloproteinases (MMPs), angiopoietin 2, and stromal cell-derived factor 1, which induces neovascularization and tissue remodelling to ensure adequate oxygen supply to the tissue (Figure 3). HIF-1-targeted therapy is in development for use in therapeutic wound healing [12]. The increased gene expression of MMPs helps endothelial cell proliferation and migration through the formation of granulation tissue on the basement membranes. MMPs also stimulate the migration of keratinocytes by degrading the protein in cells/matrix adhesions to enhance re-epithelialization. MMPs, particularly MMP-2 and MMP-9, play an important role in angiogenesis regulation during wound healing through the activation of the proangiogenic mediators TNF-α, VEGF, and antiangiogenic mediator, which degrade the basement membrane and extracellular matrix components to ensure removal of the damaged tissue [13]. Nevertheless, the overexpression of MMPs in a chronic wound can inhibit wound healing by inhibiting new tissue formation. 

On the other hand, the oxygen-dependent production of high mitochondrial-driven adenosine triphosphate (ATP) for chemical energy generation is required in tissue regeneration. Oxygen is also involved in the adenine dinucleotide phosphate (NADPH) oxidase generation of reactive oxygen species (ROS) such as superoxide and hydrogen peroxide (H_2_O_2_). These ROS regulate normal cell homeostasis and functions, upregulate cellular growth factors (vascular endothelial growth factor (VEGF) and platelet-derived growth factors (PDGF)), and induce several transcription factors that drive phagocytosis and bacteriostatic H_2_O_2_ in the cell defence response. 

ROS stimulate endothelial cell division, angiogenesis, vasculogenesis, fibroblast division and migration for the formation of collagen/the extracellular matrix, and keratinocyte proliferation and migration in tissue repair. ROS also mediate vascular constriction and dilatation through nitrous oxide (NO) following platelet exposure to the extracellular matrix. In addition, local ROS signalling for thrombus formation is crucial in the process of initial haemostasis [14,15,16]. As angiogenesis in wound healing is highly sensitive to autonomic stimuli, an adequate oxygen supply is critical for the wound tissue since the vasoactivity is high [17]. Therapeutic approaches that target improved wound tissue oxygenation can be the key to success in wound management. Hyperbaric oxygen therapy and topical oxygen are reported to enhance wound healing [18]. As an overview of its therapeutic potential for different medical conditions has been presented before [19], it is the objective of this narrative review to concentrate on the role of oxygen in oral wound angiogenesis and healing. Other means that promote angiogenesis in oral wound healing during the proliferation process, such as ultrasounds, lasers, platelet-rich plasma (PRP)/platelet-rich fibrin (PRF), and various chemical agents such as hyaluronic acid, astaxanthin, and *Centella asiatica* (*C. asiatica*), are briefly reappraised.

## 3. Oxygen Therapy

### 3.1. Ozone Therapy

Oxygen is available in two forms: the singlet oxygen (O_2_), with two molecules, and ozone, which consists of three molecules. First identified by Christian Friedrich Schönbein in 1840, ozone (O_3_) is a colourless gas with a controversial use in medical/dental therapy due to its instability, high reactivity, and toxicity [20]. More than seven decades ago, Edward Fisch used ozone therapy to successfully disinfect and heal wounds of his dental surgeries. O_3_ is available as a mixture of gases consisting of 95–99.95% oxygen and 0.05–5% pure ozone, which can be administered in a gas or liquid (water or oil) form [20,21]. O_3_ has been reported to enhance the metabolism of oxygen, induce specific enzyme processes, and activate immunological responses in the human body [18]. The enhancement of collagen type I production and the reduction of pro-inflammatory cytokines, in particular interleukin-6 and interleukin-8, following O3 exposure in human gingival fibroblasts have been documented [22].

Alzarea reported that the exposure of denture-related traumatic ulcers to 60 seconds of ozone gas was associated with better ulcer (ulcer size and ulcer duration) healing and decreased pain levels [23]. In one randomized clinical trial, ozone applied on de-epithelialized gingival grafts that were placed in the recipient bed and donor site immediately after surgery and at days 1 and 3 post-surgery showed enhanced blood perfusion units in the first postoperative week [24]. Two other clinical studies on epithelial wound healing reported cytological analyses that favoured improved healing following the topical application of ozone-treated plant oil [25,26]. 

Another study on implant surgery reported that O_3_ accelerated implant wound healing when ozonated water was irrigated at a concentration of 25 µg/mL, along with ozone gas during osteotomy, compared to a saline irrigation group [27]. Isler et al. (2018) instead compared the impact of additional topical gaseous ozone therapy on the decontamination of implant surfaces in the surgical regenerative therapyof peri-implantitis [28]. The defect site was decontaminated with saline only, while the test group received additional ozone therapy, before grafting with a mixture of concentrated growth factor (CGF) and bone substitutes. The study reported favourable outcome at a 12-month follow-up in the ozone therapy group. Despite these sporadic studies, the number of high-quality studies on the impact of O_3_ on oral wound healing is limited.

### 3.2. Hyperbaric Oxygen Therapy

Hyperbaric oxygen therapy (HBOT) is the exposure of the body to a pure concentration of oxygen (O_2_) in a pressurized atmosphere, leading to hyperoxemia and hyperoxia of the circulation and tissue [29]. The concept of this treatment first appeared in the 16th century when Henshaw, a British physician, used a “domicilium” (a chamber with compressed air) to treat chronic conditions. However, many physicians were reluctant to utilize this therapy after the first reported oxygen toxicity effect in 1789 [30]. Fast forwarding to the twentieth century, Dr. Orville Cunningham applied HBOT to treat flu-induced hypoxia successfully in the year 1921. HBOT gained popularity again after cardiac surgeon Ite Boerema (1956) at the University of Amsterdam, Netherlands successfully performed a complex heart surgery with a prolonged duration inside a hyperbaric chamber room [31]. As widely acknowledged later on, HBOT has been widely advocated for the treatment of non-healing irradiated wounds in the oral cavity since the 1980s [32]. 

Many studies are ongoing world-wide in various areas and using various conditions to explore the benefits of HBOT. Clinically, HBOT helps facilitate oxygen transfer in human tissue. It promotes angiogenesis and wound healing via HIF-1 alpha signalling activation. Following HIF-1 activation, the upregulation of NF-κB, VEGFA, SDF-1, VEGFR2, and CXCR4 have been observed. These markers are essential mediators for human skin fibroblastproliferation and angiogenesis [33]. HBOT involves patients staying in a hyperbaric chamber and breathing in one hundred percent oxygen with a pressure higher than that at sea level (>1.0 atmosphere absolute (ATA)). A minimum oxygen tension of 30 mmHg is required for normal cell division and a minimum oxygen tension of 15 mmHg is needed for fibroblast proliferation in the wound-healing process [4]. HBOT exhibits several physiological principles for how oxygen reacts under different pressure levels. The direct relationship between the oxygen concentration and the tissue diffusion gradient means that a higher concentration of oxygen increases the partial pressure of oxygen in the deeper tissues of the body. The increased atmospheric pressure also amplifies the amount of oxygen in the blood plasma, with a higher bioavailability to the tissues to a degree several times better than what can be achieved by oxygen in haemoglobin. As our knowledge of HBOT advances, it has been widely used as a therapeutic primary or adjunct treatment for a wide variety of diseases. The usage of HBOT to treat surgical/oral wounds includes the treatment of osteoradionecrosis and the enhancement of surgical flaps and grafts, implant therapy, and periodontitis.

#### 3.2.1. Osteoradionecrosis (ORN)

ORN is a common delayed complication of radiotherapy to treat head and neck malignancies, particularly following a high dosage of irradiation > 60 Gy [34]. The prevalence of ORN among the irradiated population is 5–15% [35]. However, the prevalence varies with the method of radiation delivery (3D-conformal therapy or intensity-modulated radiotherapy), the total irradiation dosage, the dose per fraction, alcohol or tobacco usage, oral hygiene, dental injury or tooth extraction, the tumour size/staging, and the patient’s age [36,37,38]. The initial concept proposed by Meyer for the development of ORN was radiation, trauma, and infection. Injury to the irradiated bone enhances microorganism invasion, causing infection. His theory became fundamental for the usage of antibiotics in the treatment of ORN [39]. In 1983, Marx proposed his new theory of hypoxic-hypocellular-hypoxia and promoted the usage of HBOT as an adjunct treatment modality for ORN besides surgery and antibiotic therapy [40]. An alternative theory of ORN is that the radiation-induced injury to the vascular system causes fibrosis and thrombosis [41]. One review instead suggested the combination of radiation-induced fibrosis and the depletion of cells in the bone because of acute inflammation, the release of free radicals, the activation of a series of growth factors, tissue damage, and the activation of chronic inflammation, which would reduce the capacity for wound healing [42]. Hence, any method such as HBOT that can enhance wound healing will become the pillar of treatment for ORN.

HBOT is highly recommended as an adjunct therapy to surgery (sequestrectomy, mandibulectomy, and reconstruction surgery) in cases of ORN [43]. Marx and colleagues (1985) reported a low incidence of ORN (5.4%) in patients treated with HBOT (2.4 ATA, 90 min, 30 sessions) and antibiotics compared to those treated with antibiotics alone (29.9%) prior to a dental extraction [32]. A systematic review in 2002, which included 14 papers, affirmed the benefit of HBOT in ORN except in one case series [44]. A Cochrane review by Bennett et al. a decade later, involving 14 trials (753 participants), suggested a similar beneficial effect of HBOT in ORN; three of the trials showed likely complete mucosa healing in ORN patients after received HBOT, and two trials discovered a less likely chance of wound dehiscence after surgery in ORN patients when given an additional session of HBOT [45]. However, a recent systematic review of 11 papers concluded that HBOT cannot be the sole therapeutic modality to replace surgery and antibiotic treatment. Its usage is comparable to antibiotics and antifibrotic medications, but with no added value [46]. On the contrary, Nabil and Samman (2011) reported a review of 19 papers that showed weak evidence that prophylactic HBOT reduces the risk of ORN in post-radiation tooth extractions [47]. Even though the utility of HBOT in ORN prevention and treatment is still controversial, it is still a beneficial adjunct therapeutic modality for those suffering from ORN, as it reduces the negative effects of radiation and improves oral wound healing, hence improving the patient’s quality of life.

#### 3.2.2. Surgical Flap/Graft

Surgical grafts and flaps are useful reconstructive procedures for the management of surgical wounds. Despite advancements in surgical techniques and perioperative care, patients still suffer from postoperative complications such as surgical site infection and anastomotic failure, particularly in the presence of comorbidities such as diabetes mellitus or post-irradiation (see above). A “surgical stress response” (non-infectious systemic inflammation with dysregulation of the neuroendocrine and metabolism systems) is one of the key factors contributing to postoperative complications besides poor wound beds, post-irradiation, vascular insufficiency, and flap necrosis [48,49]. This surgical stress response worsens the ischaemia/reperfusion activities and further debilitates tissue perfusion, especially in major surgeries performed on medically compromised patients. The clinical use of prophylactic HBOT in surgical patients has been found to improve neovascularization, reduce the post-ischemic tissue failure rate, improve flap survival, upregulate the body’s defence mechanisms, and enhance osteogenesis, besides providing bacteriostatic/bactericidal anti-inflammatory effects [50]. The overall oxidative response from HBOT-produced ROS improves the perfusion to various vital organs (heart, brain, muscles, and liver) following the ischaemia/reperfusion injury. The administration of HBOT (2.0–2.5 ATA, 90–120 min, twice daily) improves oxygen delivery from the plasma to the flap/graft tissue and enhances plasmatic imbibition as the initial stage of flap/graft healing. The further upregulation of vascular endothelial growth factors with HBOT has been shown to improve wound angiogenesis and oxygen tension in compromised flaps/grafts and change the local tissue perfusion and collagen synthesis [51].

Numerous animal studies have proven the beneficial effects of HBOT (2.0–2.5 ATA, 90 min, 14–30 sessions) on compromised graft tissues, with evidence of an increased percentage of successful transplantations, increased wound vascular regeneration, increased graft survival area, and an expedited healing time [52,53,54,55]. A review by Boet et al. in 2020 involving 13 randomized control studies (627 patients) reported that HBOT was effective in the majority (546 patients) of cases by improving at least one of the clinical outcomes (e.g., blood loss, complete healing, graft survival rate, cardiopulmonary complications, healthcare resource utilization, organ survival, etc.), while the rest of the patients did not receive any benefit or even experienced negative effects from HBOT [56]. Even though there are limitations in the heterogeneity of the research methodology, lacking blinding and sham groups, the present evidence suggests that perioperative HBOT is possibly a promising adjunct treatment for improving the outcomes of surgical flaps/grafts. Table 1 summarized the use of HBOT in various clinical applications.

#### 3.2.3. Dental Implant Therapy

For the past decade, dental implants have been commonly used for oral rehabilitation in oral cancer patients, even though the risk of implant failure in these irradiated patients remains unclear. Patients receiving radiation therapy are at a risk of reduced tissue-healing capacity with consequences of poor soft-tissue healing, osteoradionecrosis, and pathological fractures. A systematic review by Shah et al. (2017) involving 440 irradiated patients and 2250 dental implants reported a significantly lower implant failure rate in the HBOT group (9.21%) compared to the non-HBOT group (22.4%). However, confounding factors such as the total amount of radiation doses used (25Gy–145Gy), the period from the last radiation therapy to the implant placement, and the follow-up duration were not clearly stated [57]. A meta-analysis by Condezo et al. (2020) discovered contradicting results, where the researchers reported no evidence of significant implant failure risk for both irradiated patients who received dental implants and HBOT, and those without [58]. This finding is in agreement with a randomized control study, where 26 patients who were randomized to either HBOT or no HBOT showed no significant difference in their clinical outcomes in terms of implant failure, peri-implantation complications, and patient satisfaction. However, these trials had a high risk of bias and inadequate strong evidence to confirm the effectiveness of HBOT in the successful rate of dental implant placement. The effectiveness and the possible risks from HBOT should be taken into consideration when incorporating this adjunct treatment modality into dental implant procedures [59].

#### 3.2.4. Periodontal Disease

One of the most common forms of periodontal disease (gingivitis) is an inflammatory condition of the gingiva primarily caused by pathogenic bacteria such as *Porphyromonas gingivalis* (*P. gingivalis*). While this process does not involve any surgical wounds per se, the inflammatory response observed mimics those seen in oral wounds. The presence of pathogens plays an important role in the impairment of the host immune response, but is insufficient to initiate the periodontitis process [60]. Bacterial biofilm formation initiates gingival inflammation; however, the initiation and progression of periodontitis depend on dysbiotic ecological changes in the microbiome. According to the current classification of periodontal and peri-implant diseases and conditions, periodontitis is characterized by microbially-associated, host-mediated inflammation that results in the loss of periodontal attachment [61,62].

The hyperoxygenation effect of HBOT not only has deleterious effects, particularly on anaerobic periodontal pathogenic bacteria, but it also increases the ROS production. This facilitates the oxygen-dependent peroxidase system, thus enhancing the bacteriostatic/bactericidal effects of leukocytes, and at the same time supporting macrophage survival rates and VEGF production in macrophages and keratinocytes.

A review by Robo et al. (2019) focusing on the role of HBOT as a therapeutic option for patients with periodontal diseases found a significant reduction in anaerobic pathogens sub-gingivally (99%). This effect lasted up to two months post-treatment. The therapeutic effects of HBOT on gingiva, included lower gingival haemorrhage indexes, periodontal pocket reduction, reduced gingival fluid, and a reduced plaque index [63]. In treating periodontitis, hyperbaric oxygen therapy (HBOT) seems to improve clinical parameters and reduce subgingival anaerobic growth, making it a promising adjuvant in combination with standard procedures such as scaling and root planning. In a recent study, HBOT in addition to full-mouth ultrasonic subgingival debridement was administered over 10 days for the treatment of generalized severe periodontitis [64]. This modality was shown to be effective at reducing bleeding on probing and resulted in slower bacterial recolonization when compared to the control group.

Lastly, some two decades ago, the effects of HBOT in combination with near-infrared light therapy, which showed promise for delivering light deep into the tissues of the body to promote wound healing and human tissue growth, was investigated [65]. The light-emitting diodes (LEDs) were originally developed for NASA plant growth experiments in space. The study was undertaken on cells grown in culture, on ischemic and diabetic wounds in rat models, and on acute and chronic wounds in humans, all of which showed promising results, even when the LEDs were used alone. However, there was no follow-through projects on oral wound healing.

### 3.3. Topical Oxygen Therapy

A recent systematic review and meta-analysis on the use of topical oxygen therapy (tOT) suggests that it is effective and safe for chronic diabetic wound care [66]. Many different types of hydrogel dressings have been investigated, with the aim of enhancing wound healing by promoting angiogenesis while at the same time scavenging ROS. They include alginate, paramylon, fibrinogen, hyaluronic acid, silk fibroin, and carboxymethyl cellulose, among others [67,68,69,70,71]. A different approach is the administration of tOT to the affected areas in patients with peripheral arterial disease (PAD) using a local boot that delivers 100% oxygen to the wound at 1.03 atm [72].

#### 3.3.1. Hydrogen Peroxide

Taking a leaf from the medical application, the notion of applying a substance that can deliver oxygen directly to an oral wound is enticing. One source of oxygen is hydrogen peroxide (H_2_O_2_), which is an unstable oxidative disinfectant that ultimately decomposes to form water (H_2_O) and oxygen (O_2_) with resulting effervescence when exposed to glutathione peroxidase and catalase in the body. Historically, it has been widely used to disinfect wounds and is still used in some developing countries [73]. As an ROS, H_2_O_2_ was touted to promote wound healing in periodontal surgery and has been advocated as an oral rinse for many oral conditions, including acute necrotizing ulcerative gingivitis [74,75,76]. However, cases of harmful effects from hydrogen peroxide rinses have been reported [77]. The oral mucosal effects of hydrogen peroxide mouth rinses on normal volunteers were investigated almost three decades ago, with significant mucosal abnormalities observed coupled with numerous subjective complaints, resulting in its disapproval for use in oral care [78].

#### 3.3.2. Oxygen-Releasing Gel (blue^®^m)

Having said the above, products that incorporate ≤ 3% hydrogen peroxide to cleanse and promote healing of minor oral wounds and to treat gingivitis have been found to be safe [79]. They are available as an oxygen-releasing gel or mouth rinse. One product that releases oxygen and is indicated to assist in the healing process while eliminating bacteria is an oral gel (blue^®^m gel). Some studies have already proven its effectiveness.

A clinical study compared the effects of an oxygen-releasing oral gel (blue^®^m gel) and chlorhexidine gel in the treatment of periodontitis. The group that was treated with the oxygen-releasing oral gel showed better potential in probing depth reduction. The authors emphasized that thorough sub-gingival scaling and root planning, along with adjuvant tOT, aid in reducing periodontal pockets [80]. Active oxygen in a gel form used for the treatment of periodontitis selectively eradicates the anaerobic bacteria associated with periodontitis to promote the recovery of a health-compatible oral flora [81]. These results have been corroborated by a study carried out with the objective of assessing a hydro-carbon-oxo-borate complex (HCOBc) gel’s (blue^®^m gel) metabolic activity on an in vitro model of a subgingival multispecies biofilm in comparison with chlorhexidine. The results suggested that the HCOBc complex reduced the bacterial species to a smaller number when compared to chlorhexidine during subgingival biofilm formation, but it was better than chlorhexidine in reducing the proportions of red-complex bacteria (anaerobic bacteria). Although HCOBc reduced the mature 6-day-old subgingival multispecies biofilms, it did not modify the ratio of bacterial complexes as chlorhexidine did on the biofilms [82]. The bactericidal effect of blue^®^m oral gel has also been proven in another in vitro study, in which it inhibited the growth of *P. gingivalis* similarly to chlorhexidine digluconate [83].

Regarding the healing action of the oxygen-releasing oral gel, a recent study showed that blue^®^m gel can be considered as a good alternative for a Coe-Pak dressing after gingival depigmentation, owing to its pain-reduction properties, the acceleration of wound healing, and its postoperative re-epithelialization [84].

As previously mentioned, angiogenesis is a hallmark process and plays an important role in wound healing. We know that oxygen supplementation during healing aids in the oxidative killing of bacteria, the stimulation of angiogenesis, the acceleration of the extracellular matrix formation, an increased proliferation of fibroblasts, and collagen deposition, thereby enabling faster healing. The use of blue^®^m oral gel as a tOT in histological wound healing showed accelerated healing of standardized skin wounds created surgically in rats, with increased angiogenesis and better collagen fibre formation. This study also showed, by immunohistochemical analysis, a significantly higher amount of VEGF in the group that received the gel, with a slow and continuous release of oxygen [85].

The source of the slow and continuous release of oxygen in blue^®^m products comes from components present in the formula: honey (enzyme glucose oxidase) and sodium perborate. When these components come into contact with tissue fluids, they convert into H_2_O_2_ at low concentrations (0.003 to 0.15%) [86]. Oral gel is the product that has the greatest amount of oxygen release, so its main actions are healing and bactericidal.

Topical oxygen therapy has been reported to have an antioxidative effect on the treatment of peri-implantitis [87] (Figure 4). Peri-implantitis is a plaque-associated pathological condition occurring in the tissues around dental implants, and is characterized by inflammation in the peri-implant mucosa and the subsequent progressive loss of the supporting bone. Bacterial biofilm formation is considered a principal aetiological factor [88]. Studies on the treatment of peri-implantitis have revealed that anti-infective treatment strategies are successful at decreasing soft-tissue inflammation and suppressing disease progression [88]. However, a more recent review on the surgical treatment of peri-implantitis reported that the existing clinical, radiographic, and microbiological data do not favour any decontamination approaches and fail to show the influence of any particular decontamination protocol on surgical therapy [89]. On a positive note, H_2_O_2_ has the advantage of not causing significant titanium corrosion, and the corrosive properties it induces are reversible [90].

The blue^®^m company advocates topical oral oxygenation therapy (TOOTh) guidelines for the management of peri-implantitis. According to this guideline, a small amount of blue^®^m oral gel in a disposable 2.5 mL syringe should be applied in the pocket around the implant (Figure 5). The application of the gel is indicated as a chemical decontamination agent due to its bactericidal action. The home regimen includes brushing two times with blue^®^m toothpaste, rinsing the mouth three times (1 min each) with blue^®^m mouthwash, and 2–4 interdental applications of blue^®^m gel at the implant site.

The effect of blue^®^m mouthwash on oral surgical wounds has demonstrated a positive influence on tissue healing by reducing pain and the post-surgical inflammatory process [91]. A human keratinocyte cell line demonstrated a greater proliferation rate when exposed to lower concentrations of blue^®^m mouthwash [92]. Previously, one systematic review showed that H_2_O_2_ mouthwashes were not able to prevent plaque accumulation when used as a short-term mono-therapy. In contrast, the results of one study indicated that it reduced gingival redness when used as a long-term adjunct therapy [93]. A more recent systematic review, however, reported that H_2_O_2_ mouthwashes have the potential to affect plaque accumulation and gingivitis [94].

Another approach to deliver oxygen is by using toothpaste as a carrier. The anti-plaque and anti-gingivitis efficacy of blue^®^m toothpaste has already been scientifically proven by a clinical study carried out by Cunha et al. [95]. They demonstrated that toothpastes containing active oxygen and lactoferrin have comparable anti-plaque and anti-gingivitis efficacies with triclosan-containing toothpastes.

Recently, oxygen nano-bubble water has been tested for wound healing in an animal model, with the oxygen-rich liquid shown to enhance the ischaemic wound-healing process [96].

**Table 1 bioengineering-09-00636-t001:** Summary of the use of oxygen therapy to induce angiogenesis in oral would healing.

Method	Concentration/Examples	Mechanism of Actions	Oral Conditions	References
Ozone	95–99.95% oxygen and 0.05–5% pure ozone (gas, water, or oil form)		Ulcers, gingival graft surgery, peri-implantitis	[23,24,25,26,27,28]
Hyperbaric	Hyperbaric chamber and breathing in one hundred percent oxygen with a pressure higher than that at sea level (>1.0 atmosphere absolute (ATA)).	Based on concept theory of ”hypoxic-hypocellular-hypoxia”	Osteoradionecrosis	[34,35,36,37,38,39,40,41,42,43]
	2.0–2.5 ATA, 90–120 min, twice daily	Improve neovascularization	Enhance outcomes of surgical flaps and grafts	[48,49,50,51,52,53,54,55,56]
		Reduce the post-ischemic tissue failure rate; improve flap survival	Implant therapy	[57,58,59]
			Periodontitis	[63,64]
Topical oxygen	H_2_O_2_ blue^®^m (gel, mouthwash, toothpaste)	Upregulate the body’s defence mechanisms	Post-periodontal surgery, peri-implantitis	[74,75,76,79,84,85,86,87,91,92,93,94,95]
		Bacteriostatic/bactericidal with anti-inflammatory effects		

### 3.4. Gas Plasma Therapy

Gas plasma-stimulated wound healing has been utilized as a treatment therapy for chronic wounds. The multifaceted reactive oxygen and nitrogen species (ROS/RNS) are generated as a partially ionized gas using gas plasma technology. To date, marketed gas plasma systems are usually classified as medical devices (class IIa), able to be operated at body temperature and atmospheric pressure [97]. Plasma is generated using a noble gas, such as argon, by applying a high-frequency alternating voltage. The high kinetic energy accelerates the ionization process of the released electron. The gas flux drives the plasma and the charged argon particles to the ambient air containing oxygen and nitrogen, which eventually produces ROS/RNS [98]. Gas plasma has been appraised for its antimicrobial efficiency and has been shown to promote wound healing. In the medical field, the use of gas plasma is still a novelty. The literature available consists of experiments using animal models or in vitro studies that have yielded desirable wound-healing outcomes. The kINPen is an example of a gas plasma device that is being used for the treatment of infected skin diseases and wounds [99]. Abonti TR et al. (2016) demonstrated that a low-temperature multi-gas plasma jet was able to provide significant bactericidal effects to a human extracted tooth [100]. This study highlighted the potential of gas plasma to be used for oral wounds harbouring a variety of microorganisms. However, this method has raised concerns regarding genotoxicity, and requires an ideal-test system to obtain a better understanding of the effectors generated.

## 4. Other Means (Sound, Light, Biological, and Chemical Derivatives) to Promote Angiogenesis in Oral Wound Healing

### 4.1. Sound—Ultrasound

An ultrasound (US) is an oscillating longitudinal pressure wave with a frequency > 20 kHz that is inaudible to human ears. A therapeutic low-intensity ultrasound (LIU)/low-intensity pulsed ultrasound (LIPUS) has been prescribed for wound healing, as it facilitates the emission of sound waves without heat being generated. At ≤1 W/cm^2^, US therapy reduces pain and inflammation and accelerates the healing of both soft- and hard-tissue injuries [101]. Angiogenesis has been shown to occur following exposure to both short-wave (1-MHz) and long-wave (45-kHz) ultrasounds [102], with alterations in the osteoprotegerin/receptor activator of nuclear factor kappaB ligand (OPG/RANKL) ratio in human osteoblasts, resulting in new bone formation [103]. In addition, an US stimulates fibroblasts to secrete stronger and better-organized collagen during the proliferative phase, thus rapidly inducing wound healing [101].

The biophysical effects of ultrasonic energy are divided into thermal and non-thermal effects. Thermal effects result from the absorption of sonic energy, resulting in an increased blood flow to the target area. Non-thermal effects, on the other hand, include mast cell degranulation, resulting in the release of chemical mediators that attract neutrophils and monocytes to debride the wound as well as to promote healing [101]. As a result of both effects, US therapy has been shown to enhance wound healing in both soft and hard tissue. The short wave allows it to penetrate tissue as deep as 3–5 cm, as observed in hard-tissue injuries. Largely used to treat fractures, its stimulation effects are often seen during the soft callus formation phase. While it is especially beneficial for the treatment of chronic leg ulcers, its adoption for oral wounds is surprisingly limited, except for the treatment of osteoradionecrosis [104,105]. Additionally, promising preclinical studies have shown that US therapy may also promote peripheral nerve regeneration, which may be useful for treating injuries to the inferior alveolar and lingual nerves [106,107]. Lastly, ultrasounds have never been used to treat periodontitis, even though laboratory studies have reported that LIPUS facilitates the osteogenic differentiation of human periodontal ligament cells [108,109].

### 4.2. Light—Photobiomodulation Laser

Photobiomodulation (PBM) is a non-thermal light treatment that involves using endogenous mitochondrial chromophores to stimulate and modulate the biology of cells. The mechanism is believed to be the cytochrome C oxidase within the mitochondrial chromophores and the photoacceptors within the plasma membrane absorbing the irradiation. This leads to a cascade of processes, such as the production of adenosine triphosphate and nitric oxide, increases to blood circulation, and the release of reactive oxygen species, which activates the signalling of various pathways involved in cell proliferation, differentiation, survival, and tissue regeneration and healing, as described above [110].

Low-level laser therapy (LLLT) refers to radiation with a wavelength range of 500–1100 nm and a power of 1 mW–500 mW. It has the characteristic of a relatively low energy density, and has been used clinically to treat various diseases [111]. High-level laser therapy, such as non-ablative/non-thermal CO_2_ laser therapy (NACLT/NTCLT), also acts as an alternative to photobiomodulative lasers to enhance the healing process and provide pain control. NACLT produces a low-power action using a defocused beam through a thick layer of high-water-content, non-anaesthetic gel on the targeted area [112]. Laser therapy has shown beneficial effects on pain and oedema control, tissue regeneration and healing, inflammatory mediator modulation, and cell metabolism. At the cellular level, both PBM and NACLT enhance tissue oxygenation, improve microcirculation, stimulate mesenchymal cell growth, increase the tissue re-epithelialization rate, and enhance fibroblast/extracellular matrix proliferation. The positive effect of laser therapy on pain relief is achieved by changing the lymphocyte metabolism, reducing inflammatory mediator production, and altering nociceptor impulse conduction.

Laser therapy has been shown to be an effective treatment modality for the management of postoperative complications (pain, swelling, and trismus), neurological recovery, and wound healing [113]. The usage of NACLT in animal studies has shown promising results in socket preservation through the hastening of post-extraction socket healing while maintaining the alveolar bone height [114]. A similar positive result was reported in a randomised clinical study for lower wisdom tooth removal, where PBM (neodymium-doped yttrium aluminum garnet (YAG) laser) accelerated post-extraction alveolus healing with histopathological evidence of fewer inflammatory cells, a higher mature epithelium, and myofibroblasts within the alveolar mucosa [115].

The literature has shown that no standard or established protocol is used for either high- or low-level laser therapy. Regardless of the different wavelengths, doses, power, frequencies, extension of lesions, and LT protocols used, their effects are positive on both wound healing and pain control. Recent evidence has demonstrated that PBM and NACLT are effective for pain relief and wound healing in a single application without apparent side effects. NACLT exhibits superiority in its shorter duration of beam exposure (5–10 s) compared to LLLT/PBM (80–180s) [116,117].

The effect of lasers has been tested in comparison with ozone therapy in two studies [118,119]. The former study showed that the ozone group presented with statistically significantly smaller wounds as compared with the control, while the latter study showed that lasers and ozone applications after gingivectomy and gingivoplasty reduced the pain levels of patients and had a positive effect on the patients’ quality of life. When used in conjunction with H_2_O_2_, one study showed new bone formation and wound healing by secondary intention in sites treated with oxygen high-level laser therapy (OHLLT), a high-frequency and high-power diode laser combined with hydrogen peroxide (10 volume, 3%) [120].

Even though researchers have suggested that laser therapy is a promising and relatively safe alternative treatment option for oral wounds, as it promotes a greater amount of epithelialization in wound healing and excellent pain control, more randomized control studies should be conducted to develop a reliable and cost-effective laser therapy protocol with an adequate follow-up duration to identify the long-term benefits/risks of laser therapy.

### 4.3. Biological Stimulants—Platelet-Derived Products

A platelet concentrate or (PRP) consists of transforming growth factor (TGF), platelet-derived growth factor (PDGF), vascular endothelial growth factor (VEGF), endothelial growth factor (EGF), basic fibroblast growth factor (bFGF), hepatocyte growth factor (HGF), and insulin-like growth factor 1 (IGF-1). Amongst these growth factors, VEGF, FGF, and PDGF are the main regulators of angiogenesis.

Platelet-derived products have been reviewed by discussing the importance of growth factors and biomolecules related to angiogenesis that are present in plasmatic fractions with different concentrations of platelets, and their applications in traumatic injuries and degenerative diseases [121]. In a study comparing intra-articular knee injections with PRP and the viscosupplementation of hyaluronic acid in degenerative knee treatments, PRP was found to be better at reducing pain and improving knee function [122].

The second-generation platelet concentrate is also known as platelet-rich fibrin (PRF) [123]. The addition of PRF has been shown to enhance the angiogenic potential of bone substitute materials in in vivo and in vitro analyses [124]. Another version of platelet concentrate available is the lyophilized platelet-rich plasma (L-PRP). One clinical study reported that L-PRP enhanced soft-tissue healing with no difference in the extraction socket bony healing [125].

So far, there is only one study that co-evaluated the effects of PRP, systemic ozone, and hyperbaric oxygen treatments on intraoral wound healing [118]. For all three groups, the rate of intense wound closure was significantly higher than the control group in this animal study. PRP was found to be the most effective.

### 4.4. Chemical Stimulants—Hyaluronic Acid, Astaxanthin, and Centella Asiatica Extract

A commonly used gel scaffold, hyaluronic acid is a glycosaminoglycan responsible for stabilizing and organizing the extracellular matrix, mediating cell proliferation and differentiation, and regulating cell motility during tissue healing [126]. Barrier-forming hyaluronic acid-based mouth rinse and topical gel formulations have been tested for the treatment of recurrent aphthous ulcers [127], with those receiving the gel preparation reporting a faster healing onset. The effects of astaxanthin (ASTX), a xanthophyll carotenoid, have been tested on gingival fibroblasts in a wound-healing assay in vitro. The authors concluded that ASTX enhances gingival wound healing through its antioxidative properties, suggesting it as a promising candidate for the treatment of oral chronic wounds in patients with diabetes mellitus [128]. Lastly, herbs used in traditional herbal medicine, such as *Centella asiatica* (*C. asiatica*) extract, have been tested in animal studies and recently in a clinical trial, and have shown positive effects on dermal wound healing [129]. *C. asiatica* is an herbaceous vine belonging to the Apiaceae (or Umbelliferae) family that grows in the tropics, and its mechanism of action involves promoting fibroblast growth as well as increasing the synthesis of collagen and intracellular fibronectin [130]. In a human clinical trial, a *C. asiatica* gel has been shown to enhance the effect of 2940 nm Er:YAG laser therapy. When tested with Ageratum conyzoides L. leaves and asthaxantin [131], it was found that the combination of 10% A. conyzoides L. leaf ethanolic extract of the purple flower type, 5% *C. asiatica* L. Urb leaf ethanolic extract, and 0.1% astaxanthin provided the best wound-healing activity that could be developed as a commercial product. Its application in oral wounds, at best, has been tested in combination with a porcine acellular urinary bladder matrix, also with a favourable outcome [132].

## 5. Conclusions

To summarize, oxygen is an essential component for the maintenance of healthy tissues and for all the processes involved in wound healing, such as the oxidative killing of bacteria, collagen formation, epithelial cell migration, and the formation of new blood vessels. Hypoxia in the wound is a natural consequence of tissue injury, as wound healing demands energy for cell growth, proliferation, and angiogenesis, as well as for the removal of bacteria and debris. Furthermore, a plentiful supply of oxygen is required to maintain neovascularization and epithelialization [133]. This narrative review reappraises the use of hyperbaric oxygen and topical oxygen therapy, ultrasounds, lasers, PRP/PRF, and various chemical agents such as hyaluronic acid, astaxanthin, and *C. asiatica* to promote angiogenesis in oral wound healing during the proliferation process.

There are many types of topical management for oral wounds involving the soft tissue, from topically applied remedies such as *C. asiatica* or PRP/PRF to more advanced therapies such as the use of growth factors, bioengineered skin substitutes, and stem cell therapy. A product based on the principle of enhancing topical oxygen delivery to wounds is recently available in the market: blue^®^m. This product looks promising, owing to its ease of use and positive influence on tissue healing through the improvement of oxygenation without jeopardising the microbiome stability.

## Figures and Tables

**Figure 1 bioengineering-09-00636-f001:**
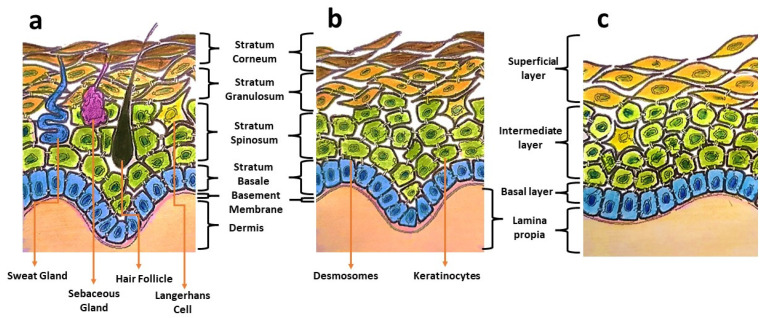
Illustrations showing differences in the components of skin (**a**), keratinized masticatory oral mucosa (**b**), and non-keratinized oral mucosal lining (**c**).

**Figure 2 bioengineering-09-00636-f002:**
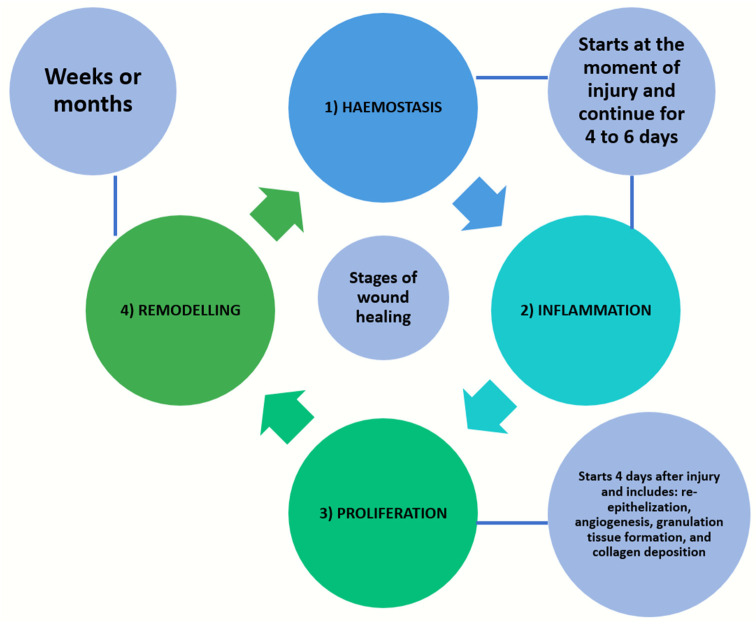
Four stages of tissue change sequences in wound healing, namely haemostasis, inflammation, proliferation, and remodelling.

**Figure 3 bioengineering-09-00636-f003:**
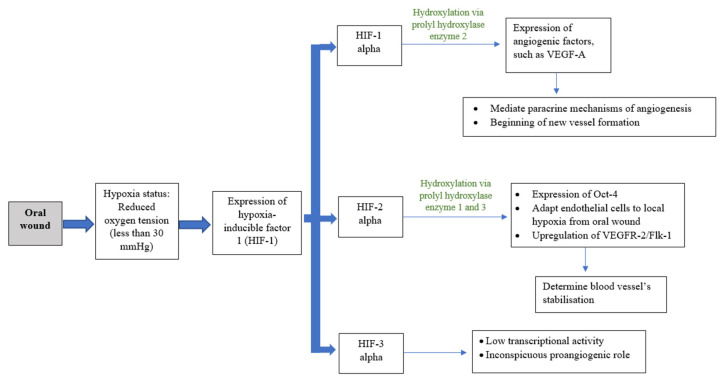
Schematic drawing showing co-relation of oxygenation and angiogenesis.

**Figure 4 bioengineering-09-00636-f004:**
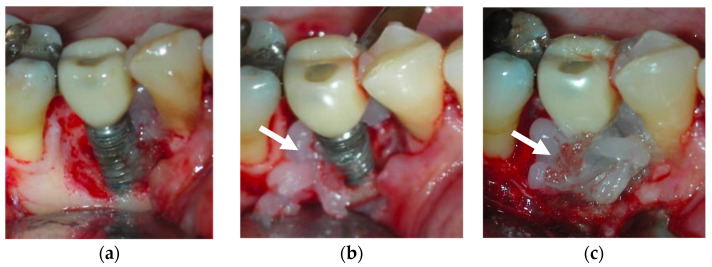
After debridement at the peri-implant site (**a**), blue^®^m was applied (**b**) and was kept on the site for 5 minutes (**c**), followed by irrigation with saline. The white arrows point to the blue^®^m gel.

**Figure 5 bioengineering-09-00636-f005:**
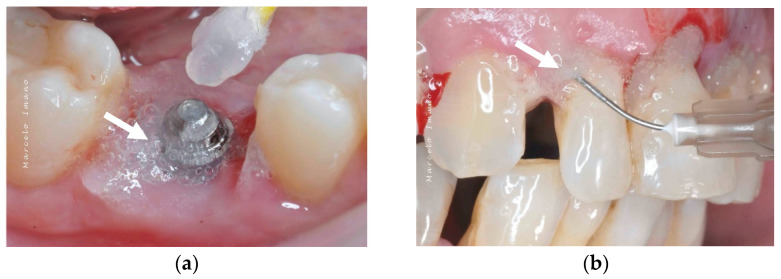
Application of blue^®^m: (**a**) as a topical gel and (**b**) using a disposable syringe in the pockets (courtesy of Dr. Marcelo Imano). The white arrows are pointing to the blue^®^m gel.

## Data Availability

The data presented in this study are available from the corresponding author upon request.

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
