# Peer review of "A Narrative Review on Means to Promote Oxygenation and Angiogenesis in Oral Wound Healing"

_bioengineering, 2022, doi:10.3390/bioengineering9110636_

Round 1
Reviewer 1 Report
Several studies have highlighted the ability of HIF-1α to stimulate a number of angiogenic factors that accelerate the wound healing process. Please describe it in section 2.1 of the manuscript.
Ø Appropriate co-relation of oxygenation and angiogenesis should be illustrated schematically.
Ø There is indecisive information about the upregulation of angiogenic markers and collagen (Collagen, being central in the regulation of several of these processes, has been utilized as an adjunct wound therapy to promote healing) after each therapy. This portion needs to be highlighted with recent work evidences.
Ø Gas plasma spurred wound healing is accompanied by regulation of tissue oxygenation should be included in treatment procedure.
Ø Please highlight Hypoxia Inducible Factor Alpha (HIF-1α) , and Matrix metalloproteinase-9( MMP-9) in context of angiogenesis in section 2.1
Reviewer 2 Report
The paper titled “A narrative review on means to promote oxygenation and angiogenesis in oral wound healing” is an interesting work that reappraises current literature on the use of light, sound, mechanical, biological and chemical means to enhance oxygen delivery to wounds. Nevertheless, this paper has some minor errors.
Minor errors:
1). They included the use of hyperbaric oxygen and topical oxygen therapy, ultrasound, laser, PRP/PRF and various chemical agents such as hyaluronic acid and astaxanthin to promote angiogenesis in oral wound healing during the proliferation process. Nevertheless, in the use of chemical agents to promote angiogenesis in oral wound healing, other agents (no only hyaluronic acid and astaxanthin) have shown a great regenerative potential in oral mucosa, among them are herbal medicines and new treatments for oral mucosa reparation based in biocompatible scaffolds. I think that is important that at least the use of Centella asiatica extract and acellular urinary bladder matrix should be included in this review. The paper Clin Oral Investig 2019; 23:2083-2095 should be included.
Reviewer 3 Report
The manuscript is a review focussed on the main methods to manage oral wounds. The literature data cover an adequate time, and the topic is very interesting. The authors reported precisely the biological assays performed to validate the substances, and methods employed to reduce these wounds.
The major issue to be revised is the style, too scholastic and immature.
A review is a critical work performed to discuss also the data already published and not merely a list of findings.
In this case, the authors are invited to personalize the manuscript, including the references in the main body harmonically. So, I suggest reorganizing the text to address my suggestions, before publication.
Reviewer 4 Report
The paper by Wei et al. summarized the current approaches to improve oral wound healing. This review is well written. I do have some comments for this paper.
1. In the introduction, it would be great to have a diagram to show the components for oral mucosa with all those cells involved.
2. In the oral wounds section, it would be great to have another sketch to show the four healing stages.
3. It would be great to summarize here why oxygen play an extremely important role in oral would healing rather than other healing process.
4. 3.3.2. Oxygen releasing gel (blue m). There is a big gap between blue and m. I believe the registered trademark symbol is missing here.
5. Figure 1. blue®m was applied following debridement at the peri-implantitis site. It would be great to have some arrows in the figure to show where is blue m if that’s possible. This comment is also applied to Figure 2.
6. It would be great if the author could expand a bit more about the importance of angiogenesis, like its molecular and functional features in oral wound healing as this paper centers on this concept.
Round 2
Reviewer 3 Report
The manuscript is ready to be published in this form